# Confocal interferometric scattering microscopy reveals 3D nanoscopic structure and dynamics in live cells

Michelle Küppers[1,2,3], David Albrecht [1,2], Anna D. Kashkanova[1,2], Jennifer Lühr[1,2] & Vahid Sandoghdar [1,2,3] ✉

Bright-field light microscopy and related phase-sensitive techniques play an important role in life sciences because they provide facile and label-free insights into biological specimens. However, lack of three-dimensional imaging and low sensitivity to nanoscopic features hamper their application in many high-end quantitative studies. Here, we demonstrate that interferometric scattering (iSCAT) microscopy operated in the confocal mode provides unique label-free solutions for live-cell studies. We reveal the nanometric topography of the nuclear envelope, quantify the dynamics of the endoplasmic reticulum, detect single microtubules, and map nanoscopic diffusion of clathrin-coated pits undergoing endocytosis. Furthermore, we introduce the combination of confocal and wide-field iSCAT modalities for simultaneous imaging of cellular structures and high-speed tracking of nanoscopic entities such as single SARS-CoV-2 virions. We benchmark our findings against simultaneously acquired fluorescence images. Confocal iSCAT can be readily implemented as an additional contrast mechanism in existing laser scanning microscopes. The method is ideally suited for live studies on primary cells that face labeling challenges and for very long measurements beyond photobleaching times.

The optical microscope was pivotal for the early development of cell biology, and although the bulk of modern biological research relies on non-imaging techniques such as biochemical assays, genetic analysis or mass spectrometry, light microscopy continues to offer highly desirable information. In fact, the recent advent of nanosciences has ushered in the exciting prospect of using microscopy for the characterization of physical phenomena such as morphology, cooperativity, diffusion, transport, and mechanical properties on the nanometer scale. Various forms of fluorescence microscopy, including super-resolution methods, have strongly steered this development, but scientists have also been intensely searching for label-free alternatives[1]. Ideally, such methods should provide dynamic three-dimensional (3D) imaging and reach single-molecule sensitivity.

The oldest form of imaging, bright-field optical microscopy, is indeed label free. The underlying mechanism of this technique is extinction of light, which can be described by the interference between the electric field of a uniform illumination beam ($\mathbf{E}_{inc}$) and the field originating from the object ($\mathbf{E}_{obj}$) at location $(x, y)$, yielding the intensity $I_{det}(x, y) \propto |\mathbf{E}_{inc} + \mathbf{E}_{obj}(x, y)|^2$ on the detector. The difference in the traveling phases of $\mathbf{E}_{inc}$ and $\mathbf{E}_{obj}$ is eliminated in bright-field microscopy, but it was later exploited in phase contrast and differential interference contrast modalities. Schemes such as reflection interference microscopy, quantitative phase imaging and digital holography[2–10] have further drawn on the role of phase to acquire quantitative information about the refractive index or thickness of a sample. Moreover, confocal microscopy has been used in the context of reflection and interference[11–15] microscopies to gain depth

[1]Max Planck Institute for the Science of Light, 91058 Erlangen, Germany. [2]Max-Planck-Zentrum für Physik und Medizin, 91058 Erlangen, Germany. [3]Department of Physics, Friedrich-Alexander-Universität Erlangen-Nürnberg, 91058 Erlangen, Germany. ✉e-mail: vahid.sandoghdar@mpl.mpg.de

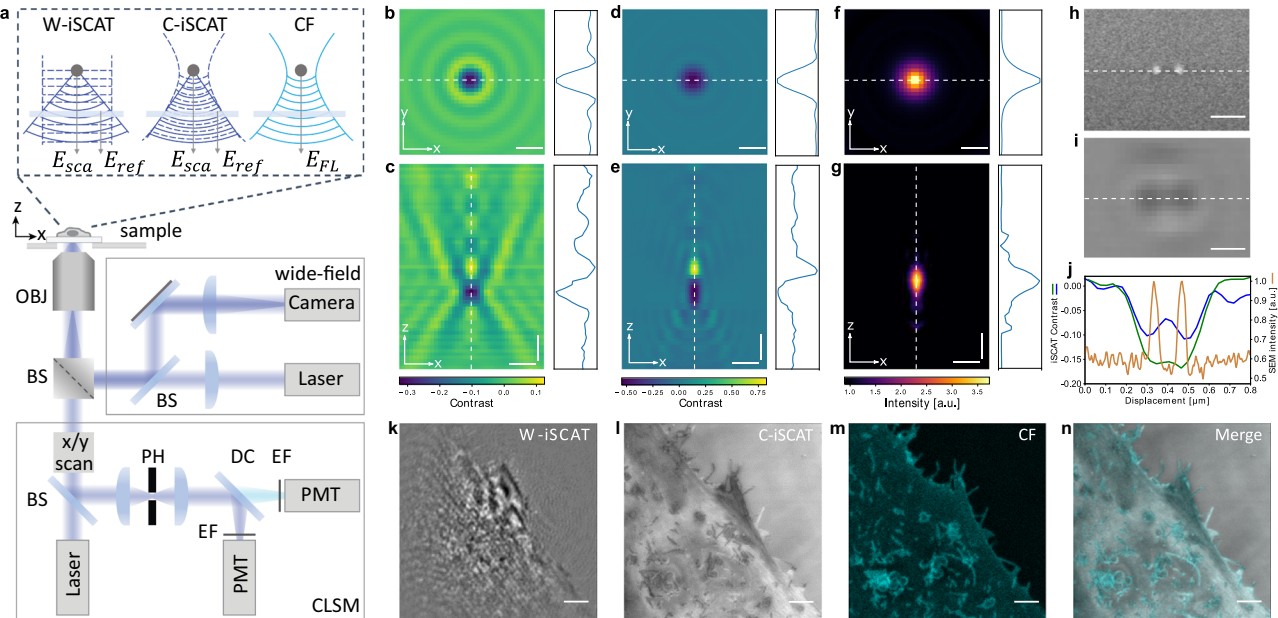

**Fig. 1 | Schematics and characterization of the optical setup. a** The main components of a confocal laser scanning microscope (CLSM), which is extended to perform iSCAT microscopy in both wide-field and confocal modes. OBJ objective, BS beam splitter, PH pinhole, DC dichroic mirror, EF emission filter, PMT photomultiplier tube. Inset: Wavefronts of laser illumination (dashed lines) and sample radiation (solid lines) for the three modalities. Lateral and axial point spread functions (PSF) of a 100 nm fluorescence-labeled polystyrene bead for wide-field iSCAT (**b, c**), confocal iSCAT (**d, e**), and confocal fluorescence (**f, g**) modalities. The focus was scanned over 4 μm in steps of 30 nm in **c, e**, and **g**. The background was accounted for in each *z* plane. Curves on the right-hand side depict the intensity profiles along the cross sections shown in each figure. Horizontal and vertical scale bars are 200 nm and 500 nm, respectively. **h** Scanning electron micrograph of a nanofabricated test sample consisting of two chromium pillars of diameter 45 nm, height 45 nm and center-to-center separation 130 nm. Scale bar is 200 nm. **i** C-iSCAT image of the sample recorded with a pinhole setting of 0.3 AU at a wavelength of 445 nm. Scale bar is 200 nm. **j** Cross sections along the white dotted lines from (**h**, orange) and (**i**, blue). The green curve also shows a cross section from a C-iSCAT image recorded with a pinhole setting of 1.2 AU. **k−n** The plasma membrane of a HeLa cell simultaneously imaged in W-iSCAT (**k**), C-iSCAT (**l**) and confocal fluorescence (**m**) modes. The plasma membrane was fluorescence-labeled with GFP-GPI. The W-iSCAT image is flat-fielded, whereas the C-iSCAT image is presented in its raw form. **n** An overlay of the images in **l** and **m**. Scale bars in **k−n** are 2 μm.

information. However, these techniques have not been optimized for sensitive detection of nanoscopic features inside the cell. Indeed, objects of concern in previous studies have been typically larger than or comparable to the wavelength of light so that the optical response of the sample is discussed in terms of *reflection* and *transmission* of light. The interaction of light with smaller objects such as vesicles, viruses, or cytoskeletal filaments should be treated in the paradigm of *scattering*[16].

In 2004, we showed that gold nanoparticles as small as 5 nm could be detected via common-path interferometric measurement of their scattering[17], introducing interferometric scattering (iSCAT) microscopy[18]. Since then, numerous studies have applied iSCAT in different illumination and detection schemes and have demonstrated single-molecule sensitivity[18]. In reflection mode, the iSCAT signal on the detector reads

$$I_{det} \propto |\mathbf{E}_{ref} + \mathbf{E}_{sca}|^2 = |\mathbf{E}_{inc}|^2 \left(r^2 + |s|^2 + 2r|s| \cos \Delta\varphi\right), \quad (1)$$

where $\mathbf{E}_{ref} = r\mathbf{E}_{inc}$ and $\mathbf{E}_{sca} = s\mathbf{E}_{inc}$ denote the electric fields of the light reflected from the sample interface and that scattered by the nano-objects, respectively (see Fig. 1a). The relative phase difference between the scattered and reflected electric fields can be expressed by

$$\Delta\varphi = \frac{4\pi}{\lambda} nz + \varphi_G, \quad (2)$$

where $n$ is the refractive index of the surrounding medium, $z$ is the axial position of the object above the cover glass, $\lambda$ denotes the wavelength of the incident light, and $\varphi_G$ stands for the Gouy phase[17,18].

The complex-valued scattering amplitude $s$ in Eq. (1) scales with the polarizability $\alpha$ of a nano-object and can be directly related to its scattering cross section. As the object becomes smaller, the contribution of the scattering intensity in this equation becomes negligible since it is proportional to $\alpha^2$[17]. We define the iSCAT contrast $C$ of a nano-object in an image according to $C = (I_{det} - I_{bg})/I_{bg}$. Equations (1) and (2) show that $C$ is modulated by the axial position of the nano-object as well as by the Gouy phase shift. To assign a contrast value that reports on $\alpha$ of a structure independently of its axial position, we also define $|C_{max}|$ as the absolute value of the maximum contrast that can be obtained from that structure.

In this article, we show that iSCAT microscopy in the confocal mode provides a remarkably powerful, sensitive, and label-free 3D imaging modality for visualization of various sub-cellular zero-dimensional (nanoparticles), one-dimensional (tubular structures) and two-dimensional features (sheets and interfaces). In particular, we show that the common-path arrangement, in which the reference beam is reflected from the cover glass, provides a simple possibility to image nanostructures up to a depth of about 4 μm. For studies at larger depths an external reference arm can be introduced. Importantly, we implemented these schemes on a commercial instrument and benchmarked our measurements with simultaneously recorded confocal fluorescence images, paving the way for their widespread adoption.

## Results
### Confocal iSCAT imaging
Figure 1a shows the schematics of an inverted confocal laser scanning microscope (CLSM, Nikon Ti Eclipse A1R) that was modified to allow for iSCAT imaging in wide-field (W-iSCAT) and confocal (C-iSCAT)

modes in addition to confocal fluorescence (CF) microscopy (see Methods). In W-iSCAT, the laser beam is focused at the back focal plane of the microscope objective to achieve a nearly-collimated illumination. To perform confocal iSCAT microscopy, we use the machinery of the CLSM to focus the incident light in the sample. The reflection at the sample-substrate interface and the light scattered by the sample features are collected by the microscope objective (numerical aperture, NA = 1.45), de-scanned via galvanometric mirrors, spatially filtered via a pinhole, and detected by a photomultiplier tube (PMT). The typical distance from an object to the reference plane sets a lower bound to the required coherence length of the employed light source.

Figure 1b displays an example of the lateral interferometric point spread functions (PSF) in W-iSCAT obtained from a fluorescence-labeled polystyrene nanoparticle that was immobilized on a cover glass. In Fig. 1c, we present the axial W-iSCAT signal of that particle obtained by scanning the focus of the microscope objective. Figure 1d–g portrays the corresponding C-iSCAT and CF PSFs for comparison. We note that the strong change of $\varphi_G$ through the focus results in an axial contrast reversal in the iSCAT PSFs (iPSF). Moreover, one observes intensity modulations away from the focus in the axial iSCAT profile, which are caused by spherical aberration (SA)[19]. Axial asymmetry, oscillations, shift and broadening in the PSF are well-known from SA in high-NA imaging[20,21] (see Supplementary Note 4). SA can be corrected to various degrees and in different ways, e.g. by adaptive optics or the use of water immersion objectives. Indeed, we previously took advantage of the resulting iPSF asymmetry caused by SA to overcome the directional ambiguity of the traveling phase for axial tracking of nanoparticles in W-iSCAT[22]. In confocal imaging, the pinhole filters the scattering that arises from the depth to a large extent, but the remaining oscillations in the iPSF (see Fig. 1e) render the iSCAT signal from two axially nearby points nontrivial. Nevertheless, as we demonstrate in the next section, the PSF of C-iSCAT allows one to image isolated 0D and 1D structures as well as interfaces. A rigorous theoretical study of C-iSCAT imaging and resolution under different conditions goes beyond the scope of this paper and will be presented elsewhere.

The expected resolution limit from Abbe's formula $\lambda/2NA$ corresponds to 152 nm for our imaging conditions at the wavelength of 445 nm. To investigate the lateral resolving power of our method experimentally, we fabricated chromium nanostructures on a glass substrate using electron beam lithography. Figure 1h displays a scanning electron microscope (SEM) micrograph of two nanopillars with a center-to-center separation of 130 nm. In Fig. 1i, we present a C-iSCAT image of this sample recorded with a pinhole size of 0.3 AU where 1 AU = $1.22\lambda/NA$ denotes one Airy unit[23]. Moreover, Fig. 1j plots cross sections from the SEM image and two C-iSCAT images registered for pinhole settings of 1.2 AU (green curve) and 0.3 AU (blue curve). We find that while the former struggles to identify the two point-like objects, the latter resolves them clearly. We note that the reference field in iSCAT imaging boosts the signal above the detector noise such that it is possible to operate with a nearly closed pinhole, whereas the pinhole is typically set at 1.2 AU[23] in CF microscopy due to a limited signal-to-noise ratio (SNR).

Confocal iSCAT microscopy can be applied to a wide range of quantitative investigations in live cells. In this work, we present a selected set of applications to demonstrate the unique advantages of C-iSCAT in terms of sensitivity and 3D imaging in live cells. To set the stage, in Fig. 1k–m, we show W-iSCAT, C-iSCAT and CF images recorded from a large section of a live HeLa cell. In these measurements, the microscope focus was placed close to the cover glass to image the basal cell membrane. We find that the W-iSCAT image exhibits a speckle pattern, which results from the interference of optical fields stemming from different regions of the sample[22,24]. The C-iSCAT counterpart, however, successfully images thin short filopodia, even

when they are located underneath the plasma membrane (see Fig. 1n for the overlay of CF and C-iSCAT images). We note that if the focus is placed further than about 4 μm in the sample, the reflection from the cover glass is filtered by the pinhole (see Supplementary Note 1). To perform iSCAT microscopy at greater depths, an external arm can be implemented. We present an exemplary solution in the Supplementary Information, including a second microscope objective and a glass-water interface to keep the same scanning and de-scanning optical path.

The outer lobes in the C-iSCAT PSF are weaker than in wide-field mode (see Fig. 1c, e) because the image in the latter is formed by the interference between a quasi-spherical and a plane wave, whereas in the former case, it results from the interference of two quasi-spherical waves (see also inset in Fig. 1a). Moreover, the detected intensity in C-iSCAT drops quadratically with the distance from the focal plane, reducing the speckle that would originate from the sample depth (see Supplementary Note 1). Another interesting phenomenon to keep in mind is that cellular structures span a broad size range, covering single proteins and sub-wavelength vesicles to extended structures such as nuclei, which are much larger than the wavelength of visible light. While Rayleigh scattering from subwavelength constituents is fairly isotropic, scattering from larger components is mostly forwardly directed[16]. This can have implications for the choice of imaging modality and its contrast quality.

## 3D reconstruction of the nuclear envelope

The nuclear envelope consists of two lipid bilayers and separates the nucleoplasm from the cytoplasm. The refractive index difference between the two media[25] allows for identifying the nucleus in many forms of imaging, including bright-field microscopy. We now show that C-iSCAT can provide an unprecedented 3D optical map of the nanoscopic corrugations of the nuclear envelope. Figure 2a displays a C-iSCAT section through the apical part of the nucleus from a HeLa cell recorded at approximately 4 μm above the cover glass. The contours of alternating contrast report on Newton rings that result from the global curvature of the nuclear envelope. In Fig. 2b, we display an optical section recorded through the basal nuclear envelope in the vicinity of the glass substrate. The blue curve in Fig. 2c plots the average of three cuts (dashed green curves) at the locations marked by the white dashed lines in Fig. 2b. A 10–90% edge sharpness analysis shows that we can localize the nuclear envelope laterally within 180 nm.

To obtain a 3D map of the nuclear envelope, we generated a C-iSCAT z-stack by recording images at different axial planes (see Supplementary Note 8). The contrast at each voxel results from the convolution of the axial iPSF profile and the optical response of the sample, e.g., caused by variations in the chemical substance, density, and morphological distribution. However, the intrinsic contrast reversal in the iPSF leads to a nontrivial axial averaging process, rendering the measured value of $C$ not directly related to structural variations for the space within a volume. The interfaces, on the other hand, provide a clear and characteristic change of signal as in confocal reflection (interference) microscopy[11–15]. To compute an iSCAT contrast $C = (I_{det} - I_{bg})/I_{bg}$ for a given structure, we need to determine $I_{bg}$. We assess this quantity by applying a low-pass Gaussian filter to the corresponding raw image so as to average over lateral signal fluctuations (see Methods and Supplementary Note 2).

The resulting interface signal can be identified in the z-stack with different analysis methods such as conventional ridge filters that detect intensity extrema[26] or by fitting a truncated oscillatory function that models the empirically measured iPSF (see Supplementary Notes 6–8). As we shall see, the high precision in this procedure allows one to resolve nanoscopic topography features. The method is particularly robust in that neither an a priori knowledge of the refractive index of the structure nor of the surrounding medium is required.

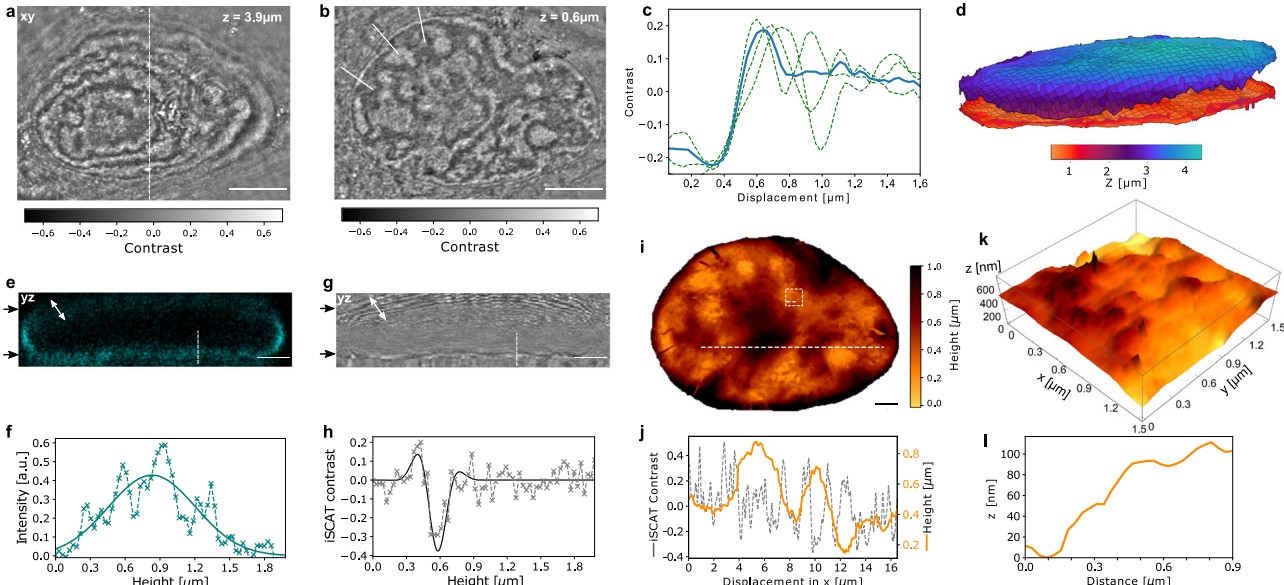

**Fig. 2 | 3D imaging of the nucleus and its envelope topography.** C-iSCAT z-stack of a HeLa cell recorded at two representative focal planes of the apical nuclear envelope at 3.9 μm (**a**) and the basal nuclear envelope at 0.6 μm above the cover glass (**b**). Each image was recorded in 1 s and was background corrected. Scale bars are 5 μm. **c** Edge sharpness analysis. Three exemplary line cuts with a linewidth of 50 pixels each (green dashed lines) and the average hereof (blue solid line). **d** 3D representation of the basal and apical nuclear envelope. **e** Projection along the yz plane (along the white dashed line as indicated in **a**) of the fluorescence signal simultaneously recorded from the nucleus, which was transfected with mCherry-laminA. **f** Axial line cut from the basal envelope (symbols) and a Gaussian fit (solid curve) along the dashed line in **e**. **g** C-iSCAT counterpart of **e**. Black arrows indicate the planes of images in **a**–**c**. Scale bars for **e** and **g** are 2 μm. Contrast color code is same as for **a**. **h** Axial line cut of the basal envelope in C-iSCAT (symbols) and a corresponding fit with the model confocal iPSF (solid curve) for height extraction along the dashed line in **g**. **i** Evaluated topography of the basal nuclear membrane after multiplane reconstruction. The height varies globally on a range of approximately 900 nm. Scale bar is 2 μm. **j** Cross section along the horizontal white dashed line in **i**. The gray dashed line shows the C-iSCAT contrast variations along this cut. The orange curve illustrates the height variations obtained from multiplane reconstruction (see Methods). **k** Rendered 3D model of a small square section shown in **i**. **l** Line cut along the dashed line in the white box in **i**, emphasizing the nanoscale morphology.

To validate our approach for reconstruction of interfaces, we imaged a nanofabricated staircase structure as well as the end of a cleaved fiber (see Methods and Supplementary Note 7).

Figure 2d displays the resulting microscopic 3D reconstruction of the basal and apical nuclear envelopes while Fig. 2e–h presents the CF and C-iSCAT data for a cut along the yz plane recorded at axial steps of $\Delta z = 30$ nm. Interestingly, the C-iSCAT axial cut through the basal envelope reveals a 10–90% edge-sharpness of 150 nm, allowing one to identify this plane with a high precision, similar to the recent results reported by Singh et al.[15]. In addition, we can apply the same ridge detection scheme to reconstruct the apical envelope at a height of about 4 μm as the objective focus is scanned upwards through the plane. At this height, the oscillations caused by SA are more prominent, but a comparison of the arrows in Fig. 2e, g, shows that the C-iSCAT and CF images are affected by SA to the same extent. The robustness of the multiplane reconstruction procedure can be compromised for the axially extended features, where the oscillations of the PSF reduce the detected contrast.

We now turn to a closer scrutiny of the image in Fig. 2b and the reconstructed surface of the basal nuclear envelope shown in Fig. 2d. Figure 2i displays a topography map of this surface, and in Fig. 2j we present the iSCAT contrast and the extracted topography for a cross section shown by the horizontal dashed line in Fig. 2i. The observed axial modulations in the order of 600 nm are consistent with previous reports from electron microscopy (EM) studies[27]. To showcase the capability of C-iSCAT for surface nano-profilometry inside live cells, in Fig. 2k we portray a small section of the square region marked in Fig. 2i as well as a cross section of it in Fig. 2l. We remark that such investigations have not been accessible to other optical methods. Our finding that the basal nuclear membrane contains nanometer-scale ripples contributes to the recent discussion on the effect of cytoskeletal forces on the nucleus[28] in the general context of mechanotransduction[29].

## Endoplasmic reticulum structure and dynamics

The endoplasmic reticulum (ER) is a dynamic membrane network consisting of tubules with a diameter of 50–100 nm and membrane sheets confined to a luminal spacing of 50 nm[30]. The ER contrast has been too weak for bright-field microscopy or holo-tomographic microscopy[31], but ER tubules have previously been visualized by phase contrast microscopy[32] and quantitative phase imaging[33]. We now show that the combination of high sensitivity and efficient background rejection in C-iSCAT enables extended label-free and 3D measurements of native ER. Figure 3a, b presents simultaneously recorded CF and C-iSCAT sections of the ER network at the periphery of a live COS-7 cell. We note that in this thin region the weakly scattering structures of the ER network are readily visible in C-iSCAT images even prior to background treatment. To acquire a quantitative understanding of the ER network, we employed an actively stabilized focus lock system and recorded time-lapse images of live cells (see Supplementary Movie 2). A large contrast of about $|C_{max}| \simeq 12\%$ in C-iSCAT images allows for efficient segmentation using common algorithms such as ridge detection based on eigenvalues of a Hessian matrix[26] and a thresholding operation.

The ability of C-iSCAT to detect a wide range of cellular nanostructures in a label-free fashion is accompanied by lack of specificity. To identify and distinguish a certain organelle or structure, one can employ a convolutional neural network (see Methods) analogous to what was recently demonstrated for bright-field microscopy[34]. To demonstrate this for the ER network, we used the CF channel as ground truth to train a conditional generative adversarial neural network (cGAN) on 2500 data sets of 256 × 256 pixels[35]. Consequently, we applied the trained cGAN to the iSCAT image in Fig. 3b. The outcome is shown in Fig. 3c, whereby the color map presents the prediction output of cGAN, which matches the CF data in Fig. 3a to within 92%.

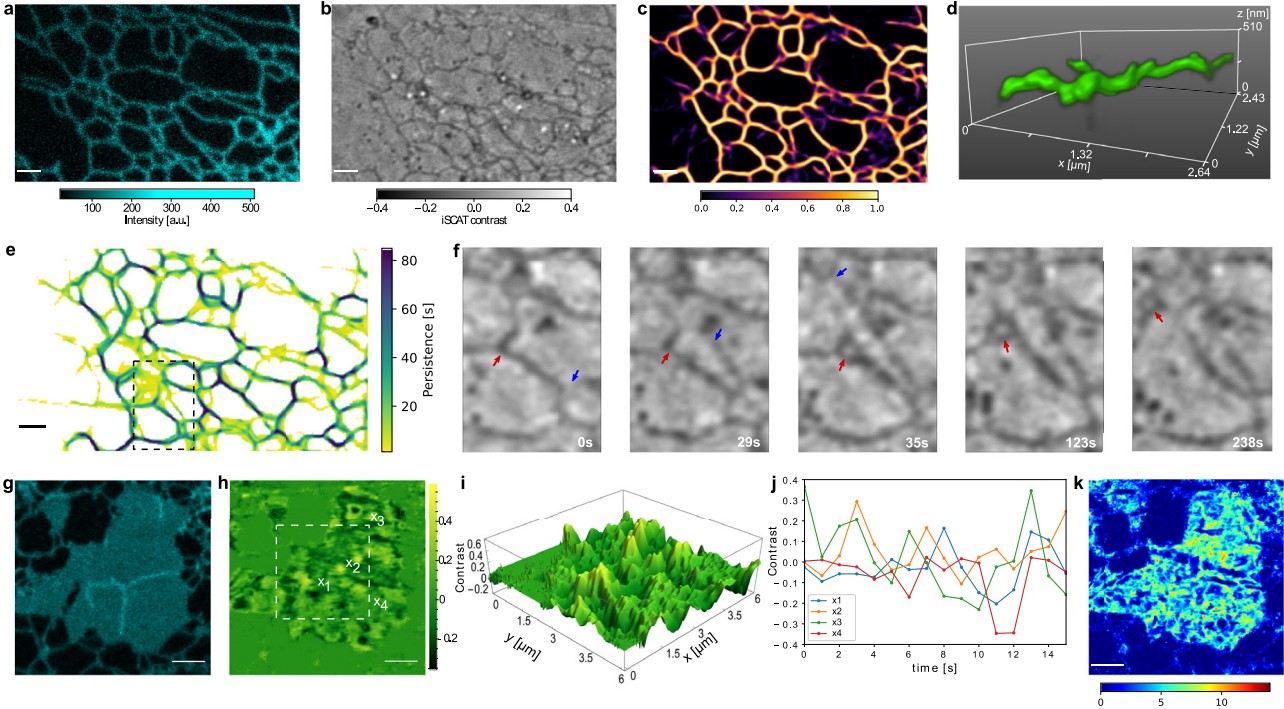

**Fig. 3 | Structure and dynamics of the endoplasmic reticulum. a** Fluorescence image of the ER network in a COS-7 cell that was transfected with ER-EGFP. **b** Simultaneously recorded background-corrected C-iSCAT image. ER tubules in the cellular periphery yield a mean iSCAT contrast of 12%. Scale bars in **a** and **b** are 1 μm. **c** Corresponding segmentation of ER after training a conditional generative adversarial network (cGAN). Color map encodes the prediction output of cGAN. **d** Rendered 3D representation of the tubular network obtained from a *z*-stack over the range of 510 nm recorded with Δ*z* = 30 nm. **e** Persistency map calculated using 340 frames over 85 s with a lag period of 5 s. The color code corresponds to the duration each pixel is occupied by a sub-structure of the tubular network. Scale bar is 1 μm. **f** Time course of the region marked by dashed lines in **e**. Formation of a ring-like nanodomain (red arrows) and ER sliding (blue arrows) are revealed. **g** Fluorescence image of an ER sheet labeled with EGFP. Scale bar is 5 μm. **h** Corresponding C-iSCAT contrast map of the segmented ER sheet. Scale bar is 5 μm. **i** Pseudo-3D contrast map of the region marked in **h** indicating axial sheet fluctuations. **j** Exemplary oscillation profiles at four positions indicated in **h**. **k** Dynamic map of the ER sheet shown in **h**. The color map denotes occurrence of contrast inversions at each pixel over a period of 16 s. Scale bar is 5 μm.

To portray the 3D complexity of the ER network, we again recorded z-stacks. We then applied a multiplane analysis based on fitting a truncated oscillatory function that mimics the empirically measured iPSF data (see Fig. 1e and Supplementary Note 9) after segmenting the ER network in each focal plane. Figure 3d shows a 3D reconstruction of a small section of those data, whereby the apparent thickness of the tubule represents the uncertainty in localizing its center.

An important task in modern microscopy is to investigate the dynamics and function of biological processes. Achieving this goal is challenging in fluorescence studies because low count rates restrict the imaging speed while photobleaching limits long observation times. In Fig. 3e, we present the dynamics of the ER network over 1.4 min in a persistency map with a lag time of 5 s, distinguishing highly dynamic (yellow) from static (dark blue) regions. Here, we identify network domains as well as tubules connected in three-way junctions, which remain stable for longer than one minute. Furthermore, we observe a continuous remodeling of the ER network by fusion and fission. Temporal snap shots of an area marked in Fig. 3e are shown in Fig. 3f, revealing ring closure fusion events and the formation of new ER segments (see arrows and Supplementary Movie 2). We remark that such quantitative analyses of ER architecture and dynamics have only been recently reported based on fluorescence microscopy[36]. We expect the label-free 3D imaging capability of C-iSCAT to become a great asset in this research area.

As shown in Fig. 3g, ER sheets can be easily detected in fluorescence microscopy, but further information about the out-of-plane structure of this thin organelle is not accessible. The simultaneously recorded iSCAT counterpart of the image, on the other hand, reveals large contrast variations up to ±40% (see Fig. 3h). In Fig. 3i, we depict a different representation of a portion of the same data to portray a more intuitive view of the modulations. Assuming negligible refractive index fluctuations across the sheet, our data indicate nanoscopic displacements along the optical axis. This finding, which has not been previously accessible to optical investigations, is in line with the results of EM studies, which also detect 3D variations in ER sheets[37].

The topographic features of ER sheets have recently been predicted to entail temporal dynamics[38], but they have not been within reach for experimental verification. Our iSCAT data are, indeed, able to image these dynamics. Figure 3j displays temporal traces of the iSCAT contrast for four different locations marked in Fig. 3h, demonstrating oscillatory motion on the order of seconds. In Fig. 3k, we plot a map of the number of oscillations for the entire ER sheet presented in Fig. 3g, h recorded over 16 s (see Supplementary Movie 3). Interestingly, these data also exhibit fluctuation nodes (i.e., zero motion), which we attribute to locations of contact with the cytoskeleton.

## Imaging microtubules

Microtubules (MTs) are cytoskeletal filaments that form a hollow cylinder with an outer diameter of ≈25 nm comprising tubulin[39,40]. MTs provide a scaffold and molecular track for intracellular structure and transport[41]. Label-free microscopy methods have imaged and studied MT dynamics in vitro[42–45], but imaging MTs in live cells has relied on fluorescence imaging[46].

To minimize the effect of the cellular background in C-iSCAT imaging, we adjusted the refractive index of the medium by adding the contrast agent iodixanol, which is known to be compatible with live

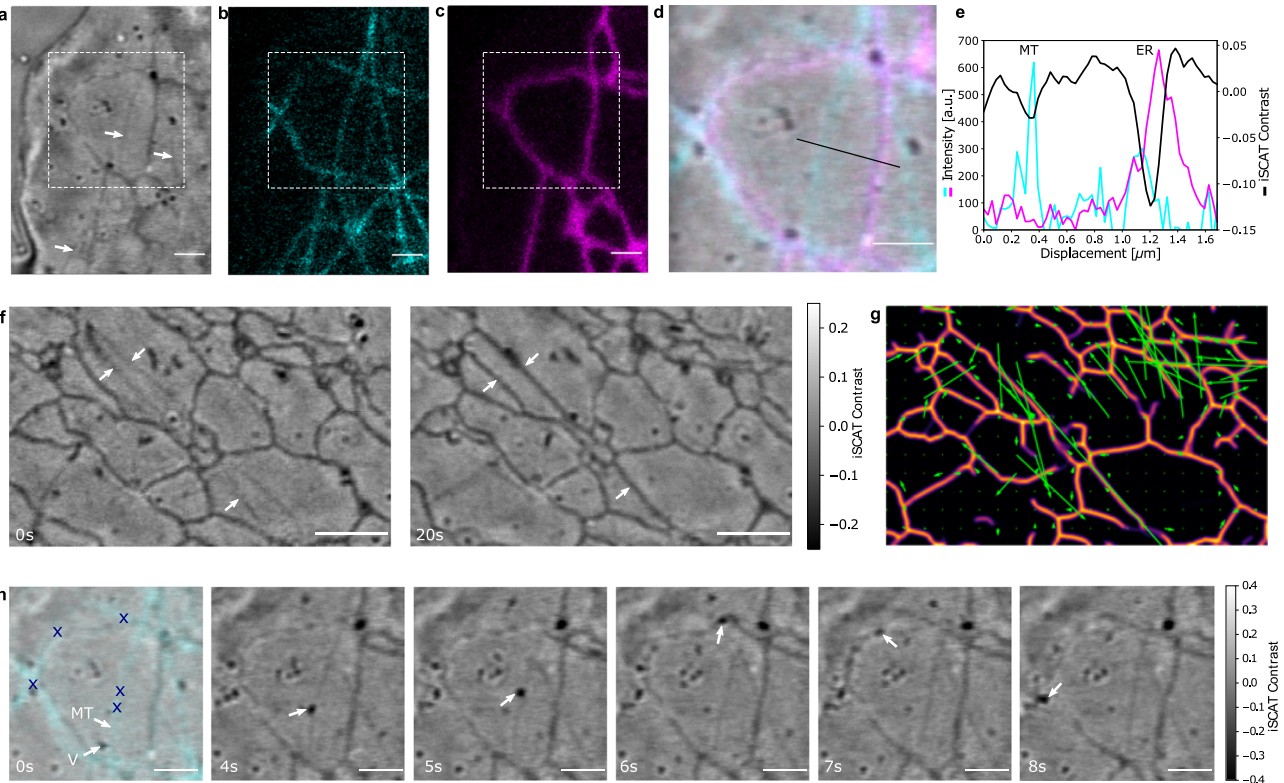

**Fig. 4 | Ultrahigh sensitivity in imaging microtubules in live cells. a** Exemplary raw C-iSCAT image of the periphery of a COS-7 cell showing vesicles, ER network and MT. Corresponding CF images of MTs labeled with mEGFP-tubulin (**b**) and ER tubules labeled with CytERM-mScarlet (**c**). Scale bars are 1 μm. **d** Overlay of **a–c** for a region marked in **a**. Scale bar is 1 μm. **e** A cross section along the cut in **d**, displaying the C-iSCAT (black) and fluorescence signals of MT (cyan) and ER (magenta). The focal plane was adjusted to be able to extract the maximum negative contrast of both MTs and ER tubules. **f** Exemplary raw C-iSCAT image from the periphery of a COS-7 cell at $t = t_0$ and $t = t_0 + 20s$. The white arrows indicate the configuration of two visible MTs in between the ER tubular network. ER tubules undergo motor-dependent "sliding" along the length of the MTs. Scale bar is 2 μm. **g** Optical flow map depicting ER sliding and dynamics of ER nanodomains. Direction and length of the green arrows indicate the direction and amplitude of motion of the ER network. **h** Time course over 8 s shows the position of the previously identified MT and a vesicle being transported along it. Symbols mark the consecutive locations of the vesicle. Scale bar 1 μm.

imaging[47]. We chose the refractive index of the medium to match the cytoplasm $n_{cyt} \approx 1.365$. The resulting decrease in reflectivity at the interface between the cover glass and the medium also leads to an overall increase in contrast. Figure 4a shows a C-iSCAT image from the periphery of a COS-7 cell, and Fig. 4b, c presents CF images obtained by simultaneous labeling of MT and ER. A close examination of the three images and of the overlay of a smaller region depicted in Fig. 4d shows that we are able to detect and image MTs. The iSCAT and fluorescence signals along the line cut in Fig. 4d are presented in Fig. 4e. The mean iSCAT contrasts of ER and MT amount to $|C_{max}| \simeq 12\%$ and 2%, respectively. To our knowledge, this is the first report of sufficient sensitivity and resolution in label-free imaging of MTs inside live cells.

Simultaneous label-free imaging of the MT with other organelles allows one to investigate their interactions and dynamics. For example, the two raw C-iSCAT snap shots in Fig. 4f disclose ER tubule extension along MTs (see white arrows), a process termed ER sliding[48]. An optical flow map in Fig. 4g illustrates the ER dynamics, whereby the direction and length of the green arrows indicate the direction and amplitude of motion. In addition, the analysis of the ER tip pointed to a directed trajectory with a mean velocity of about 2 μm/s, suggesting motor-dependent ER extension along a MT (see Supplementary Movies 6 and 7)[41,48].

MTs also serve as network for intracellular transport of vesicles. Revisiting Fig. 4a, we note the presence of a fairly high density of diffraction-limited spots, which are consistent with the footprint of nanoparticles. Figure 4h presents snap shots that reveal active transport of a nanoparticle with $|C_{max}| \simeq 35\%$ over 8 s (see white arrow). The individual positions marked in the first panel highlight the directional

change of this cargo switching between two MTs (see Supplementary Movies 4 and 5). Here too, label-free 3D imaging of transport processes can be expected to enable more sophisticated studies than has previously been reachable.

## 3D tracking of clathrin-coated pits

Isolated nanoparticles occur in several different contexts within the cell. Figure 5a presents a C-iSCAT image of the cell periphery, where many nanoparticles can be identified. The majority of the detected particles exhibits a contrast of $|C_{max}| \simeq 20\%$. By keeping the focal plane at a constant position, we recorded a time series for this region and used a radial variance transform (RVT) to localize the nanoparticles[49].

Figure 5b depicts the trajectories of each nanoparticle, revealing that most entities experience a root mean squared displacement in the order of 200 nm over the time of observation. The color map encodes the absolute contrast change along each nanoparticle trajectory. Contrary to the particles close to the cell body (yellow), those in the cell periphery often do not experience contrast reversal and remain within the focal plane as indicated with a small $|C_{max} - C_{min}|$ value (purple). To identify the origin of these nano-objects, we expressed clathrin light-chain fused to mCerulean in live cells and recorded CF images (see Methods). Figure 5c presents the overlay with Fig. 5a. Co-localization events between the C-iSCAT and CF signals let us conclude that a large fraction (about 65%) of the detected nanoparticles consists of clathrin-coated pits, which mediate endocytosis and generate small vesicles of 60–120 nm in the process[50]. In Fig. 5d, we display the temporal evolution of clathrin-coated pits in a small region from Fig. 5c. The direct relation between the optical phase variations and the axial

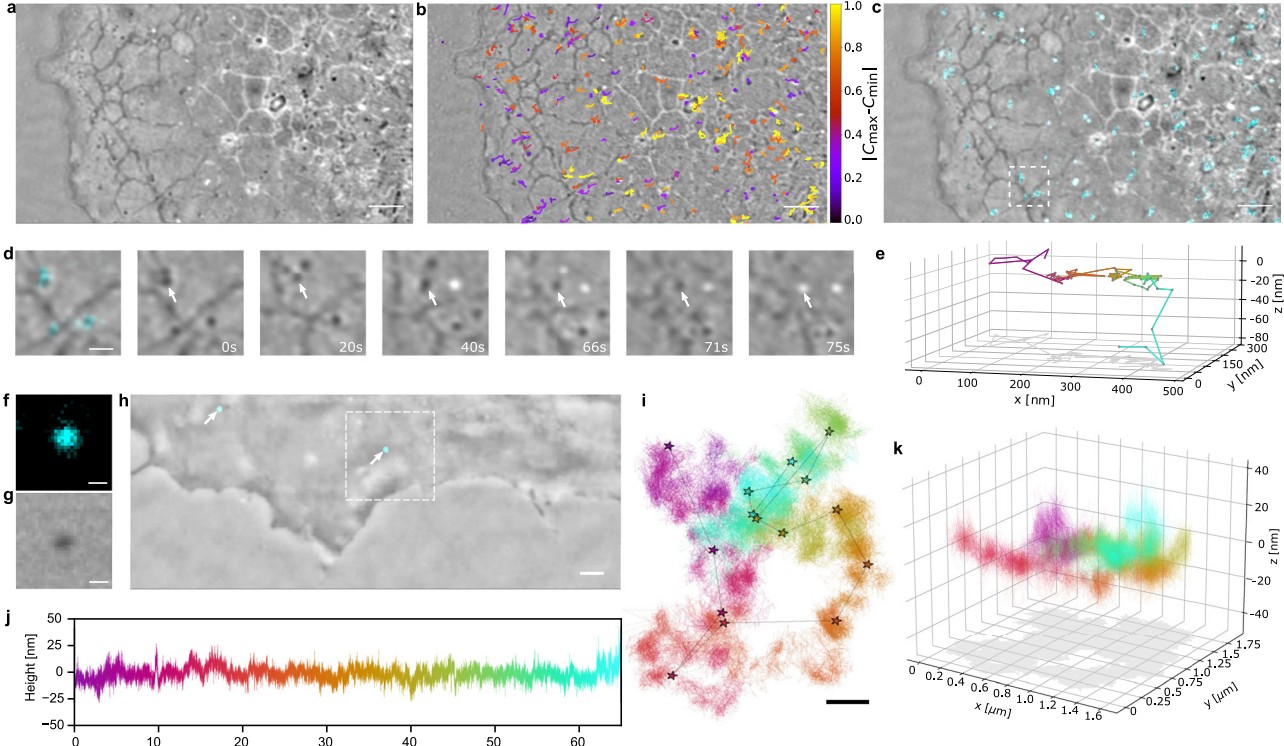

**Fig. 5 | 3D tracking of clathrin-coated pits and SARS-CoV-2 on the plasma membrane of a live cell. a** Exemplary raw C-iSCAT image of the periphery of a COS-7 cell showing vesicles, ER network and MTs. Scale bar is 2 μm in **a**–**c**. **b** Overlay of single particle trajectories of vesicles detected and tracked in **a**. Color map encodes the absolute value of the contrast change normalized for each trajectory. **c** Overlay of **a** with CF of clathrin labeled with mCerulean3 (cyan). **d** Close-up of **c** including clathrin-coated pits. Time course over 75 s shows contrast inversion of a clathrin-coated pit indicated with a white arrow. Scale bar is 500 nm. **e** Extracted 3D trajectory of the clathrin-coated pit marked in **d**. Color map encodes the temporal evolution starting from $t = 0$ s (magenta) to $t = 75$ s (cyan). CF (**f**) and C-iSCAT (**g**)

images of a single SARS-CoV-2 particle on a bare cover glass. Scale bar is 200 nm. **h** Merge of the C-iSCAT and CF (cyan) signals of labeled SARS-CoV-2 particles on the plasma membrane at the periphery of a COS-7 cell. Scale bar is 1 μm. **i** Color-coded data show a highly dense 2D trajectory of a diffusing SARS-CoV-2 particle obtained in W-iSCAT from the field of view indicated by the white dashed square in **f**. The star symbols represent the trajectory of the same event recorded simultaneously in the CF channel. Scale bar is 250 nm. **j** Extracted height displacement of the SARS-CoV-2 particle over time obtained from W-iSCAT images. **k** 3D representation of the trajectory depicted in **i**. Colors in i and k follow the same map as in **j** and encode time over the course of 65 s at a frame rate of 1 kHz.

position of a nanoparticle (see Eqs. (1) and (2)) allows one to attribute small changes in the iSCAT contrast to $z$ displacements[18]. Figure 5e shows an example of a resulting 3D trajectory for a clathrin-coated pit (see white arrow in Fig. 5d), disclosing a rapid axial translocation within 3 s, which is consistent with an endocytosis event.

## Combining W-iSCAT and C-iSCAT for high-speed virus tracking

The intrinsic scanning process in confocal microscopy limits the imaging speed. We now show in a last case study that the combination of C-iSCAT and W-iSCAT allows for both cellular imaging and high-speed nanoparticle tracking. We demonstrate this combination for the particularly important application concerning the interaction of a SARS-CoV-2 virion with a cell, which spans time scales ranging from sub-second to days. Viruses ideally lend themselves to our studies because they can provide a substantial contrast in iSCAT[51,52].

Figure 5f, g displays exemplary images of one virion on a cover glass in the CF and C-iSCAT channels, respectively. In Fig. 5h, we present an overlay of CF and C-iSCAT images containing two SARS-CoV-2 virus particles in the periphery of a COS-7 cell (see Methods). Once the virus particle was identified in the CF image, we could follow it using iSCAT microscopy with a contrast of $|C_{max}| \simeq 7\%$ even after it was photobleached. We note that although this contrast is sufficiently high to detect a virus in the iSCAT channel, its combination with fluorescence microscopy offers a particularly efficient and robust imaging strategy.

To increase the imaging speed beyond that of confocal scanning, we added the W-iSCAT mode at 1 kHz acquisition speed (see Methods). Figure 5i plots a 2D trajectory containing more than 60,000 locations, visualizing the diffusion of a single virus particle on the plasma membrane over approximately one minute. For benchmarking and comparison purposes, we also present a trajectory (see star symbols) extracted from the simultaneously recorded CF images, which clearly falls short in terms of spatial precision and temporal resolution. In Fig. 5j, we display the relative height of the virus particle extracted from its iSCAT contrast, depicting axial displacements in the order of tens of nanometer. As in the case of the clathrin-coated pit presented in Fig. 5d, e, the contrast variations were small to avoid phase ambiguities. Using this information, we generated a 3D trajectory that is plotted in Fig. 5k. Interestingly, there are clear regions of higher occupancy and regions which are revisited multiple times.

## Discussion

We have shown that confocal iSCAT microscopy provides access to label-free structural and dynamical investigations of various nanoscopic constituents in live cells. Bypassing the need for fluorescence labeling brings about several important advantages. First, sample preparation is simplified. Second, all organelles are accessible at the same time, providing immediate insight into their dynamic correlations. Third, lack of photobleaching permits unlimited observation times of cellular activities. Lastly, the absence of photochemistry reduces photodamage. In addition, the interferometric nature of

C-iSCAT provides high-resolution access to nanoscopic displacements in the axial direction. Furthermore, the iSCAT modality is readily compatible with simultaneous fluorescence imaging to allow for benchmarking and co-localization studies.

The main goal of our work has been to develop and establish an interferometric confocal microscopy mode for quantitative studies of intracellular nanoscopic phenomena. Intriguingly, our proof-of-principle measurements have already led us to several unprecedented imaging achievements in cell biology. We have shown the potential of C-iSCAT for nano-profilometry on intracellular surfaces such as the nuclear envelope. Moreover, we presented the morphology and dynamics of the ER network such as ring formation and sliding along MTs. In addition, we unveiled axial ER sheet oscillations, which had been postulated but not yet observed. Furthermore, we demonstrate the first label-free live cell imaging of MTs and cargo transport as well as clathrin-coated pits undergoing endocytosis. Finally, we introduced the combination of confocal and wide-field iSCAT for simultaneous live cell imaging and high-speed single particle tracking in the timely example of SARS-CoV-2 diffusion on the plasma membrane.

Many of these investigations were carried out towards the periphery of cells, but C-iSCAT is also applicable to measurements in deeper cellular regions either in the common-path mode presented in this work or by adding an external interferometer arm as discussed in the SI. The achievable signal-to-noise ratio depends on the background dynamics and the desirable temporal resolution. The measured contrast is dictated by the polarizability of the nanostructure under study, which is in turn determined by the refractive index, size, and shape of the nanostructure. Moreover, the contrast is affected by the strength of the reference, which should be monitored (see Supplementary Note 1). Careful characterization of these parameters would thus merge conventional quantitative phase imaging and C-iSCAT at the nanoscopic scale.

Every microscopy method aspires to reach single-molecule sensitivity. Fluorescence microscopy achieved this goal in the early 1990s[53], but background fluorescence continues to make its application to live cells quite challenging. Similarly, iSCAT microscopy of single emitters and proteins has been demonstrated in well-controlled media such as clean coverslips[54,55] or artificial lipid membranes[56]. Although that performance is currently not achievable within cells, we showed that C-iSCAT can visualize and track sub-wavelength entities such as vesicles, clathrin-coated pits and viruses in live cells. Combining these measurements with diffusional analysis[57] promises to provide further quantitative information about the size and composition of cellular nano-objects. Moreover, increasing the laser power can help capture faster dynamics and smaller entities although it should be kept in mind that at some point absorption would lead to photodamage. We emphasize that the measurements presented in our current work were performed at laser powers in the order of 5 µW with a typical dwell time of about 1 µs per frame per diffraction-limited spot. This corresponds to a light dose well below what is considered to cause photodamage[58] (see Supplementary Note 12 for phototoxicity checks).

A decisive technical advantage of C-iSCAT over other label-free techniques is that it can be readily implemented in existing confocal laser scanning microscopes. Indeed, C-iSCAT was successfully implemented in a spinning disk microscope after the submission of our current manuscript[59,60]. Our methodology holds promise for a wide spectrum of fundamental studies on the origin of ER dynamics, recruitment and regulation of MTs, interaction of motor proteins and specific cargo, quantitative understanding of nuclear mechano-regulation, clathrin-mediated endocytosis, and more[50,61]. Considering the range of studies that are enabled by the method and its compatibility with simultaneous fluorescence imaging, we expect C-iSCAT to become a routine technique in live cell imaging.

## Methods

### Experimental setup

The schematics of the experimental arrangement used to perform iSCAT microscopy in confocal and wide-field configurations is shown in Fig. 1a. W-iSCAT was performed with a custom-built addition to a commercial confocal laser scanning microscope (Nikon Ti Eclipse A1R). The output of a white-light laser (NKT Photonics, SuperK Extreme exr-15) was coupled into a single-mode fiber (SM450 P3-460Y-FC-2, Thorlabs). For wide-field illumination, the output was focused at the back focal plane of a Nikon Plan Apo 100x Oil 1.45 NA objective. To couple the wide-field beam path into the microscope axis of the stand, we added a 30:70 (R:T) beam splitter to the base plate of the microscope stand. The detection path is arranged for the focal plane of the imaging lens to coincide with the back focal plane of the objective. This results in the scattered field to be focused and the reflected reference field to be planar on a sCMOS camera (MV-D1024-160-CL-8, Photonfocus).

For confocal iSCAT imaging, we replaced the main dichroic mirror of the microscope commonly used for the separation of the excitation and detection beam paths by a 20:80 (R:T) beam splitter. With this minor modification the light of a 445 nm diode laser is focused (by the same objective as W-iSCAT) and raster scanned with the built-in galvanometric mirrors across the sample. Both the incident light reflected at the interface between the cover glass and the imaging medium and the light scattered by the sample are collected by the objective, de-scanned via the mirrors, spatially filtered via the pinhole and detected by a photomultiplier tube (PMT). The dichroic mirror and emission filter in front of the PMT are chosen such that the wavelength of the illumination beam lies within the bandpass of the filters, i.e., 450/50 nm in order to measure the intensity of the total light field. In comparison to W-iSCAT, this configuration detects the interference of two quasi-spherical waves.

### Sample preparation

Cell culture and fluorescence labeling: Human HeLa cells (DSMZ) were cultivated in Dulbecco's modified Eagle's medium (DMEM) supplemented with 10% fetal bovine serum (FBS) and 2 mM Glutamax. African green monkey kidney COS-7 cells (DSMZ) were cultivated in DMEM supplemented with 10% heat-inactivated FBS. Cells were maintained in a humidified incubator supplemented with 5% $CO_2$. Routine mycoplasma tests were negative. Cells were seeded on 35 mm glass-bottom cell culture dishes (ibidi) at a density of $12–60 \times 10^3$ cells/cm$^2$ at least one day prior to imaging.

Fluorescent fusion protein constructs were transiently transfected using lipofectamine 3000 according to the manufacturer's instructions with 0.5–1 µg plasmid DNA per dish. Cells were imaged 18–48 h after transfection. Plasmids used were pCytERM-mScarlet-H-N1, Addgene 85067 or ER-EGFP, subcloned in-house (endoplasmic reticulum); mCherry-LaminA-C-18, Addgene 55068 (nuclear lamina); GFP-GPI, kind gift from Jason Mercer Lab (plasma membrane); mCerulean3-Clathrin, Addgene 55408 (clathrin-coated pits); "pmEGFP a tubulin IRES puro2b", Addgene 21042 (microtubules). Glass-bottom dishes were optionally plasma cleaned to reduce background fluorescence of unspecifically binding lipophilic dyes.

Sample mounting and imaging buffer: Cells were washed with pre-warmed (37 °C) PBS and the medium was exchanged to phenol red-free FluoBrite medium, Leibovitz' medium or Leibovitz' medium (all Thermo Fisher) with 30% v/v iodixanol (Sigma Aldrich) prior to imaging. Live microscopy was performed at 37 °C in humidified air supplemented with 5% $CO_2$ in a commercial microscopy incubation chamber.

SARS-CoV-2 inactivation and labeling: To avoid the need for measurements under high-security laboratory conditions, alternative model systems have been used for tracking virus interaction with live cells. Commonly used systems for studying SARS-COV-2 include

pseudoviruses made by incorporating spike protein (S) into virus-like particles derived from other viruses such as HIV-1[62] or conjugation of S proteins to inorganic nanoparticles such as quantum dots[63]. Neither approach captures the diversity seen in virus progeny during infection with SARS-CoV-2. To visualize the motion of a SARS-CoV-2 virion on the cell membrane, we purified and chemically inactivated viruses harvested from the supernatant of infected cells. We also labeled these viruses with the lipophilic dye R18 to allow for CF imaging. Virions were then pipetted on stage to live cells.

## Dielectric nanoparticles for characterizing the iPSF

To characterize the interferometric point-spread function (iPSF) of the setup, we imaged single polystyrene fluorescence-labeled beads. 35 mm cell culture dishes (ibidi) were coated with poly-L-lysine (Sigma) for 5 min at room temperature (RT), aspirated, and rinsed with deionized water. 505/515 FluoSpheres (40 nm and 100 nm, Thermo Fisher) were diluted at 1:200 in deionized water and incubated for 10 min at RT to bind to the cover glass.

## Chromium nanopillars for assessing the resolution

To assess the resolution of C-iSCAT microscopy in a textbook manner, we prepared two nano-objects at a close separation. The clean cover-slips were first coated by a 50 nm thin Cr layer and 200 nm thick negative tone electron beam resist (AR-N 7520, Allresist GmbH). Electron exposure using 100 kV acceleration in an electron beam lithography machine created the desired structures in the resist film, which was transferred into the Cr layer by reactive ion etching employing a $ClO_2$ chemistry. Finally, the resist was removed using an oxygen plasma leaving the 50 nm tall Cr pillars. In this fashion, we produced chromium nanopillars of varying size and distance on a glass substrate. For optical imaging, the sample was immersed in the same medium as that used for live cell imaging (containing iodixanol, see above).

## Background correction

To assign a contrast to the C-iSCAT images and thus acquire quantitative information about the sub-cellular structure and dynamics, we first have to determine the background level $I_{bg}$. For fields of view including regions of bare cover glass, one can readily assess the reflected intensity $I_{ref}$, which can serve as the background. For a more general approach, we approximate a background by applying a low-pass filter to blur the surrounding of the structure of interest. Here, we apply a Gaussian kernel of varying width and determine an empirical cut-off kernel size to maintain the visibility of the feature of interest while blurring the environment. In our applications, we found the appropriate kernel size to lie between 16 and 32 PSF FWHM (see Supplementary Note 2).

## Segmentation via cGAN

For quantitative studies of intracellular structures, it is helpful to segment the information obtained from C-iSCAT frames in order to distinguish the features of interest from other variations. Segmentation can be implemented with different methods such as ridge detection based on eigenvalues of a Hessian matrix[26] followed by a thresholding operation. However, these techniques require user-defined criteria, which might be difficult to optimize. Recent advances in machine learning provide convenient methods for addressing this issue in a more robust manner. Here, we used a conditional generative adversarial network (cGAN). As shown in Fig. 3b, the ER visibility in C-iSCAT is high enough to be detected in a single frame even without background correction. Hence, we have chosen this structure for an exemplary demonstration of the segmentation scheme.

We trained a cGAN on a set of videos in which the ER was also labeled fluorescently. We used data sets from 10 different cells. The size of the video ranged from 512 × 512 px to 1024 × 1024 px with each

pixel corresponding to 30 nm. Only the first few frames from each data set were used in order to ensure fluorescence data with high SNR. Next, a Conditional Adversarial Network[35] was trained on the data as follows. One of the 76 images was selected at random. It was re-scaled along $x$ and $y$ by two random factors drawn from a normal distribution with width 0.1 centered around 1. A 256 × 256 section was selected at random. Then a random rotation by 0, 90, 180 or 270 degrees and vertical mirror was applied. This was repeated 2500 times. The model was trained on this data set over 50 epochs.

## Multi-plane reconstruction

Equations (1) and (2) show that the iSCAT contrast changes as one scans the focus of the microscope through a structure. This is also seen in the empirically measured axial iPSF of our microscope presented in Fig. 1e. To obtain a 3D reconstruction of a structure, we first record a z-stack. Next, the signal in each plane is background corrected to account for the variations in contrast that result from changes in the reference beam intensity (see Supplementary Notes 2, 6 and 7). In a further step, the images can be segmented to eliminate spurious background. Then a suitable approach is employed for analyzing the axial profile of the iSCAT contrast at each voxel. Here, one can use a conventional ridge detection or fit a truncated oscillatory function that models the measured iPSF (see Supplementary Note 6). The achievable axial precision depends on the SNR of the contributing image planes. The accuracy in the laboratory frame can be affected by a number of phenomena, including spherical and chromatic aberrations.

## Single-plane reconstruction

The interferometric nature of iSCAT imaging allows one to acquire information about the axial dynamics of nanoscopic features even without scanning the focus, i.e., by imaging a single plane. In other words, assuming that the refractive index remains constant, the contrast variations can be interpreted as relative $z$ displacements for a moving object. This approach is particularly applicable to nanoscopic displacements, where the resulting phase change does not cause contrast reversal and phase wrapping ambiguities.

Clathrin-coated pits: In the case study of clathrin-coated pits, contrast variations were used to render their 3D motion. To overcome the phase ambiguity in this approach, we used the knowledge that when clathrin pits on the plasma membrane are internalized, they can only move inward in the axial direction. We found a mean maximum contrast of $|C_{max}| \simeq 20\%$ for clathrin-coated pits based on observations in several cells. This value was used to calibrate the maximum distance traveled in $z$ for a full contrast inversion, which amounts for our wavelength to 110 nm.

ER sheets: For a continuous structure such as a membrane sheet at an arbitrary depth, phase ambiguities in the approach described above make a unique assignment of topography impossible. Nevertheless, to gain insight about sheet fluctuations, we rendered a pseudo height map based on the iSCAT contrast to represent the axial motion within the sheets without a quantitative assignment. To get around this complication and obtain the height variation, future efforts could acquire additional information via a multiple-wavelength approach for a single focal plane or apply the above mentioned multi-plane approach with a single wavelength. The temporal resolution of the latter approach is limited by the scanning speed. This restriction can, in turn, be alleviated by reducing the scanned size of the field of view (FoV).

## Single-particle tracking

Individual nanoparticles can be tracked if they can be localized separately in each frame. While PSFs in fluorescence microscopy can be approximated by 2D Gaussian profiles, the richer structure of iPSFs in iSCAT render this approach less robust. To address this issue, we recently introduced a method based on the radial variance transform

(RVT) to localize nanoparticles in W-iSCAT[49]. In our current work, we applied RVT also to C-iSCAT videos. For tracking of the converted data, we employed Trackpy[64] and set a lower threshold of at least 25 localizations per trajectory.

## Combination of C- and W-iSCAT in the same FoV

In order to enable simultaneous imaging with W-iSCAT, C-iSCAT and CF via the same objective, a careful alignment of the beam combiner is in order. Since the maximum FoV in C-iSCAT (about 100 µm) is much larger than that of W-iSCAT (about 10 µm), one can easily identify the illumination profile of W-iSCAT in the confocal scan. To do this, we de-scanned the reflection of the wide-filed illumination and detected it on a second PMT of the C-iSCAT channel after spectral separation. For fine alignment, we determined the relative positions of the two FoVs with a calibration sample. Here, we immobilized 100 nm fluorescent nanoparticles (505/515, Thermo Fisher) on a cover glass and deduced the transformation necessary to transfer both into one coordinate system.

## Animating 3D C-iSCAT datasets with 3D script

To visualize our results in a more intuitively accessible fashion, we animated the 3D reconstructed structures of the ER tubular network. We employed 3D script as a plugin for ImageJ, which enables animation of z-stacks via a customized rendering algorithm[65]. In order to apply this method to our data, we first created virtual z-stacks from the obtained height maps. For the final rendering of the iso-surfaces, we used a Gaussian filter with $\sigma = 2$ in the $z$ direction.

## Statistics and reproducibility

The presented data show representative recordings of our experiments. We have repeated all measurements several times with different rounds of sample preparation. We obtained similar results for all experiments and they showed low statistical variation. For the characterization measurements with 100 nm fluorescence-labeled polystyrene beads (Fig. 1b–g and Supplementary Figs. 3 and 4) we imaged at least 50 individual particles from various FoVs. The chromium nanopillars were fabricated as a matrix with varying diameter and distance between each set of two pillars, where each combination of diameter and distance consists of a field of 4×4 replicates. For the experiments labeling the plasma membrane of COS-7 cells (Fig. 1k–n) we acquired more than 10 data sets, taken from more than three different cell culture dishes on different preparation days. For the experiments imaging the nuclear envelope in 3D (Fig. 2 and Supplementary Figs. 2, 11–14) we acquired more than 20 data sets, taken from more than 6 different cell culture dishes on different preparation days For the experiments imaging the endoplasmic reticulum (Fig. 3 and Supplementary Figs. 15 and 16) we acquired more than 50 data sets, taken from more than 10 different cell culture dishes on different preparation days. For the experiments imaging the microtubules (Fig. 4) we acquired more than 10 data sets, which were taken from more than 5 different cell culture dishes on different preparation days. For the experiments imaging clathrin-coated pits (Fig. 5a–e) we acquired more than 10 data sets, taken from more than 3 different cell culture dishes on different preparation days. For the experiments imaging SARS-CoV-2 (Fig. 5f–k) we acquired more than 30 data sets, which were taken from more than 3 different cell culture dishes. For the fiber tip experiments to determine spherical aberration more than 20 measurements were performed with one tip (Supplementary Figs. 5 and 8) For the staircase experiments to determine the height extraction we measured more than 10 steps from one fabricated sample (Supplementary Fig. 10) For the fiber tip experiments to demonstrate the depth extension via an external reference more than 20 measurements were performed with one tip (Supplementary Fig. 18) For the experiments determining the phototoxicity of our method (Supplementary Fig. 19) we measured more than 25 cells, which were imaged on two separate days.

## Reporting summary

Further information on research design is available in the Nature Portfolio Reporting Summary linked to this article.

## Data availability

Considering the large size of the individual raw videos, the datasets are available from the corresponding author V.S. on request. Requests will be answered within 3 weeks. Source data are provided with this paper. The in-house subcloned plasmid will be shared via mail upon request made to the corresponding author.

## Code availability

Wide-field iSCAT data acquisition was performed using a custom cam-control software developed in our lab (https://github.com/SandoghdarLab/pyLabLib-cam-control)[66]. The confocal data were acquired using the Nikon software NIS-elements AR 5.02.03. The main data analysis was written in Python 3.6 and used standard Python packages as well as scikit-image v.0.18.1, trackpy v.0.5.0, keras v.2.3.0, scikit-learn v.0.24.1. Several different analyses were performed on the different types of datasets during the presented study. The code for each analysis can be provided together with the raw datasets from the corresponding author. Training and segmentation of the ER is based on the publication on Image-to-image translation with conditional adversarial networks[35] and was adapted from the tutorial https://machinelearningmastery.com/how-to-develop-a-pix2pix-gan-for-image-to-image-translation/. 3D surface plots were carried out in Mathematica 12.3 and Matlab R2020b. Animation of 3D data sets was performed with "3D script" as a plug-in for ImageJ2[65].

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

## Acknowledgements

The authors are grateful to Jan Renger for the fabrication of the chromium test sample, Eduard Butzen for SEM imaging of the test samples, Simone Ihloff for cell culture, Benjamin Schmid (Optical Imaging Center Erlangen, OICE) for help with the 3D visualization of the SI videos, Maksim Schwab for machining the customized beam combiner, Tobias Utikal for help with the implementation of the external interferometer arm and Alexey Shkarin for help with automated readout of the piezo position for fiber tip measurements. We also thank Alexandra Schambony, André Gemeinhardt, Martin Blessing, Luis Morales, Hisham Mazal, Mahdi Mazaheri, Richard Taylor, Cornelia Holler and Philipp Tripal (OICE) for fruitful discussions and helpful comments on the manuscript. We also acknowledge support from the Institute of Virology at the University Hospital of Friedrich-Alexander University in Erlangen. This work was financed by the Max Planck Society.

## Author contributions

M.K. implemented the combination of C-iSCAT, W-iSCAT and CF microscopy modalities. M.K., D.A., J.L. performed the experiments. M.K. and A.K. analyzed the data. D.A. and J.L. prepared the live cell materials. V.S., M.K., D.A. and A.K. prepared the manuscript, which was discussed and commented by all authors. V.S. conceived and supervised the project.

## Funding

## Competing interests

The authors declare no competing interests.
