## [Peer Review File · Nature Communications]

Confocal Interferometric Scattering Microscopy Reveals 3D Nanosopic Structure and Dynamics in Live CellsThis manuscript has been previously reviewed at another journal that is not operating a transparent peer review scheme. This document only contains reviewer comments and rebuttal letters for versions considered at *Nature Communications*.

REVIEWER COMMENTS

Reviewer #1 (Remarks to the Author):

I would like to thank the authors for revising the manuscript. I was impressed by the details that the authors have provided. Importantly, the authors clearly answered to the raised questions, particularly on the novelty. The overall quality of the revised manuscript is very high, and my previous concerns were all adequately addressed. Thus, I strongly support the publication of the manuscript.

Reviewer #2 (Remarks to the Author):

Reply to authors resubmission:

1. C-iSCAT vs reflectance confocal microscopy.

The authors state in their general remarks that "Exploiting scattering of nanostructures and emphasizing its role as opposed to reflection from larger structures is indeed a central aspect of our work" in comparison to previous work. They also point out the role of interference. Here, I would like to point out that scattering and interference are bases for previous reflection microscopy implementations that successfully imaged nano structures. For example, Simmert et. al., (PMID: 29877486) and Mahamdeh et.al., (PMID: 30044498) imaged microtubules and gold nano-particles in in vitro assays using traditional implementations of reflection microscopy (with partially coherent light). The contrast generated in the images was a result of interference between the scattered light and the back reflected illumination. In fixed cells, Keith and Farmer (PMID: 17481347) imaged individual microtubules using reflection-enhanced backscatter confocal microscopy. Again, contrast is generated from interference between scattered and reflected reference laser light. The exciting and novel aspect of this work is that the imaging was done in live cells with sufficient contrast and deeper axial scanning capabilities because, as the authors mentioned, interaction of light with sub-cellular nanoscopic features was analyzed in a quantitative matter. Nonetheless, the authors text edits helped to clarify the confusion all reviewers had about the novelty of the method.

2. Height maps.

The authors answered my questions about height maps generation and provided data and explanation of how they measured the axial resolution.

3. Single and multiplane reconstructions.

I believe the manuscript would've benefited from fast axial scans of a small FOV but I understand the authors inability to do so due to COVID pandemic impacting hardware supply and delaying vendor services.

4. Photodamage controls.

The authors addressed my concern about photodamage; they performed control measurements that showed no negative impact on cells due to the 445nm laser used in C-iSCAT.

Conclusion

The authors improved the manuscript; both text and data. They also answered my questions and addressed concerns related to photodamage. I believe that the technique is useful and has many potential applications in biology. Thus, I recommend publishing this manuscript in Nature Communications.

Reviewer #3 (Remarks to the Author):

The authors had addressed most of my comments sufficiently and made corresponding modifications in the revised manuscript. The demonstrations of visualizing various cell nanostructures are exciting and of immediate interests to the cell biologists. A thorough

characterization of spatial resolutions (instead of precision) and quantitative interpretation of C-iSCAT image data are still not possible due to the lack of iPSF and the complication of spherical aberration. Nevertheless, the manuscript clearly shows the potential of C-iSCAT for cell biology. I would be happy to recommend its publication after the following issues being clarified.

1. As being pointed out by referee #1, deconvolution would help interpret quantitatively the C-iSCAT image data. Current deconvolution strategies for coherent imaging require the data of complex fields, including the complex signal fields and the complex PSF (see for example, ref.5 <https://doi.org/10.1038/nphoton.2012.329>). In fact, previous works on deconvolution with a complex point source readily treat the problem by considering scattering (beyond reflection and transmission), so the statements in Line 35-38 are incorrect and thus need to be revised. (Line 35-38: 'Indeed, objects of concern in previous studies have been typically larger than or comparable to the wavelength of light so that the optical response of the sample is discussed in terms of reflection and transmission of light. The interaction of light with smaller objects such as vesicles, viruses, or cytoskeletal filaments should be treated in the paradigm of scattering [16].')

2. The difficulty of deconvolution of C-iSCAT image data largely stems from the fact that C-iSCAT does not produce the quantitative complex scattered fields. In other words, C-iSCAT is not quantitative phase imaging (QPI) yet. The authors are recommended to state this and discuss about future development of QPI in C-iSCAT.

3. A laser with a relatively short coherence length (~150 micron) was used in C-iSACT. Many laser scanning confocal microscopes are equipped with laser diodes that exhibit a coherence length of a few millimeters. The authors need to comment on the effect of coherence length on the image quality of C-iSACT, ideally with a demonstration.

4. Based on eq (2), a phase difference of 90 or 270 degrees occurs when objects lying at certain heights relative to the coverglass, generating no interference contrast. Is this a fundamental limitation, or are there any methods to restore the contrast for those objects?

5. Line 242: '...a large fraction of the detected nanoparticles consists of clathrin-coated pits,...' Please specify the fraction.

6. There is a relevant work on C-iSCAT by using spinning disk confocal scanning unit which shall be added in the reference and discussed [<https://doi.org/10.1364/OE.471935>].

7. The statement in Line 97 that transmission microscopy is less sensitive and has a lower axial resolution is biased. As noted by the authors (stated in the response to my comment #3), the sensitivity of interference detection is set by the number of detected photons. Small nanoscopic cell structures scatter equally in the forward and backward directions, so the detection sensitivity should be the same in transmission and in reflection. The fact that most transmission microscopes reach a lower sensitivity is because they are operated at a lower incident intensity (otherwise the illumination saturates the detector). The incident power of transmission microscopy can be increased by attenuating the transmitted beam in the detection path (as being done in several interference microscope modalities, including phase contrast, DIC microscopy and others). As a result, transmission interference microscopy can also be very sensitive (in fact, it has been demonstrated that iSCAT is equally sensitive in transmission and in reflection [<https://doi.org/10.1039/C8NR06789A>]). There is also no evidence that transmission microscopy has a lower axial resolution. I suggest the authors to either remove this statement or explain it explicitly in order to avoid misunderstanding.

8. The statement in Line 100 is misleading: 'when imaging smaller sub-cellular structures, it is advantageous to carry out iSCAT measurements in a back-scattering geometry to reduce the background.' It implies that the main difficulty of visualizing nanoscopic cell structures is the background created by the large cell structures. I believe that the issue of scattering background in cell imaging is generally more complicated. For example, as evident in the data of this manuscript, the backscattering geometry picks up strong interface reflections that preclude the detection of nanostructures nearby (this is why it is difficult to observe nanostructures above or below the highly reflective nuclear membranes in C-iSCAT). The trouble of reflective background

can be effectively resolved by detection in transmission. In sum, the pros and cons of forward versus backward detections are more complicated than what being stated, and they should not be oversimplified.

RESPONSE TO REVIEWER COMMENTS

Reviewer #1 (Remarks to the Author):

I would like to thank the authors for revising the manuscript. I was impressed by the details that the authors have provided. Importantly, the authors clearly answered to the raised questions, particularly on the novelty. The overall quality of the revised manuscript is very high, and my previous concerns were all adequately addressed. Thus, I strongly support the publication of the manuscript.

Our reply:

We are grateful to the reviewer for his/her valuable feedback which has helped improve our manuscript.

Reviewer #2 (Remarks to the Author):

Reply to authors resubmission:

1. C-iSCAT vs reflectance confocal microscopy.

The authors state in their general remarks that "Exploiting scattering of nanostructures and emphasizing its role as opposed to reflection from larger structures is indeed a central aspect of our work" in comparison to previous work. They also point out the role of interference. Here, I would like to point out that scattering and interference are bases for previous reflection microscopy implementations that successfully imaged nano structures. For example, Simmert et. al., (PMID: 29877486) and Mahamdeh et.al., (PMID: 30044498) imaged microtubules and gold nano-particles in in vitro assays using traditional implementations of reflection microscopy (with partially coherent light). The contrast generated in the images was a result of interference between the scattered light and the back reflected illumination. In fixed cells, Keith and Farmer (PMID: 17481347) imaged individual microtubules using reflection-enhanced backscatter confocal microscopy. Again, contrast is generated from interference between scattered and reflected reference laser light. The exciting and novel aspect of this work is that the imaging was done in live cells with sufficient contrast and deeper axial scanning capabilities because, as the authors mentioned, interaction of light with sub-cellular nanoscopic features was analyzed in a quantitative matter. Nonetheless, the authors text edits helped to clarify the confusion all reviewers had about the novelty of the method.

2. Height maps.

The authors answered my questions about height maps generation and provided data and explanation of how they measured the axial resolution.

3. Single and multiplane reconstructions.

I believe the manuscript would've benefited from fast axial scans of a small FOV but I understand the authors inability to do so due to COVID pandemic impacting hardware supply and delaying vendor services.

4. Photodamage controls.

The authors addressed my concern about photodamage; they performed control measurements that showed no negative impact on cells due to the 445nm laser used in C-iSCAT.

Conclusion

The authors improved the manuscript; both text and data. They also answered my questions and addressed concerns related to photodamage. I believe that the technique is useful and has many potential applications in biology. Thus, I recommend publishing this manuscript in Nature Communications.

Our reply:

We are grateful to the reviewer for his/her valuable feedback which has helped improve our manuscript.

Reviewer #3 (Remarks to the Author):

The authors had addressed most of my comments sufficiently and made corresponding modifications in the revised manuscript. The demonstrations of visualizing various cell nanostructures are exciting and of immediate interests to the cell biologists. A thorough characterization of spatial resolutions (instead of precision) and quantitative interpretation of C-iSCAT image data are still not possible due to the lack of iPSF and the complication of spherical aberration. Nevertheless, the manuscript clearly shows the potential of C-iSCAT for cell biology. I would be happy to recommend its publication after the following issues being clarified.

Our reply:

We thank the reviewer for carefully reading our manuscript and for an overall positive assessment of our work. We address the individual comments below. All changes in the main text of the manuscript are highlighted in orange.

Reviewer #3:

1. As being pointed out by referee #1, deconvolution would help interpret quantitatively the C-iSCAT image data. Current deconvolution strategies for coherent imaging require the data of complex fields, including the complex signal fields and the complex PSF (see for example, ref.5 <https://doi.org/10.1038/nphoton.2012.329>). In fact, previous works on deconvolution with a complex point source readily treat the problem by considering scattering (beyond reflection and transmission), so the statements in Line 35-38 are incorrect and thus need to be revised. (Line 35-38: 'Indeed, objects of concern in previous studies have been typically larger than or comparable to the wavelength of light so that the optical response of the sample is discussed in terms of reflection and transmission of light. The interaction of light with smaller objects such as vesicles, viruses, or cytoskeletal filaments should be treated in the paradigm of scattering [16].')

Our reply:

We make a point by emphasizing the difference between the response of very small (well below a wavelength) objects and those that are larger than a wavelength. Considering that there is no severe boundary between the two regimes, we have indeed chosen our wording very carefully: We qualify our assessment by the use of the word "typically" in the above-mentioned passage. Moreover, the passage should be read and interpreted in its context, discussing digital holography, quantitative phase imaging and reflection interference microscopy. We continue to be of the opinion that the great majority of the previous efforts have not focused on the detection of subwavelength nano-objects. For example, the authors of Ref 5. also study neurons, diatoms and bacteria on a larger scale. This work does not emphasize intracellular structures, that are comparable to or smaller than the wavelength of visible light.

We respectfully disagree that our statements are incorrect.

Reviewer #3:

2. The difficulty of deconvolution of C-iSCAT image data largely stems from the fact that C-iSCAT does not produce the quantitative complex scattered fields. In other words, C-iSCAT is not quantitative phase imaging (QPI) yet. The authors are recommended to state this and discuss about future development of QPI in C-iSCAT.

Our reply:

As we emphasize in the manuscript, C-iSCAT and QPI share their fundamental interferometric signal, but as the referee points out, iSCAT does not yield immediate quantitative information on complex scattered fields. Indeed, iSCAT aspires to complement and extend the parameter space that is accessible to QPI, rather than competing with it. We believe we have adequately acknowledged the applicability regime of iSCAT in the discussion and outlook section by stating: "The measured contrast depends on the polarizability of the nanostructure under study, which is in turn determined by the refractive index, size, and shape of the nanostructure." We have now

slightly expanded this passage to read "Careful characterization of these parameters would merge conventional quantitative phase imaging and C-iSCAT at the nanoscopic scale."

Reviewer #3:

3. A laser with a relatively short coherence length (~150 micron) was used in C-iSCAT. Many laser scanning confocal microscopes are equipped with laser diodes that exhibit a coherence length of a few millimeters. The authors need to comment on the effect of coherence length on the image quality of C-iSCAT, ideally with a demonstration.

Our reply:

The user has some freedom in choosing the coherent length of the laser. A strict requirement is for the coherence length to be longer than the path difference between the reference and scattering object. Considering that the working distance of the high-NA immersion objective is around 130 μm , a coherence length of 150 μm is sufficient. Of course, light sources with higher coherence lengths can also be used. However, it should also be kept in mind that unnecessarily long coherence lengths could introduce spurious speckle for instance from dust, dirt or other imperfections on optical elements. On the other hand, the pinhole will help suppress such effects to some extent. Ultimately, the choice of the coherence length should be optimized by the user or manufacturer according to the application and the available equipment.

We now include a comment in the section on confocal iSCAT imaging.

Reviewer #3:

4. Based on eq (2), a phase difference of 90 or 270 degrees occurs when objects lying at certain heights relative to the coverglass, generating no interference contrast. Is this a fundamental limitation, or are there any methods to restore the contrast for those objects?

Our reply:

As the referee points out, the contrast of the central lobe of the iSCAT PSF can vanish for a nano-object at certain heights above the cover glass and for a given position of the microscope objective. We point out, however, that the central lobe is also accompanied by side lobes/rings with alternating contrast (see Fig. 1 and Refs. 19, 50). In other words, even when the contrast of the central lobe vanishes, the side-lobes continue to provide valuable information about the axial position of the object. In C-iSCAT the side lobes are suppressed compared to the WF-iSCAT (see Fig. 1), but they remain nonzero. Furthermore, if one scans the objective in the axial direction, the null contrast of the central lobe transitions to a finite value due to the change in the Gouy phase. Thus, a 3D volume scan and multiplane reconstruction provide additional information to avoid any ambiguities.

Reviewer #3:

5. Line 242: '...a large fraction of the detected nanoparticles consists of clathrin-coated pits, ...'
Please specify the fraction.

Our reply:

We thank the reviewer for bringing up this question. We confirm that about 65% of all detected particles showed fluorescence if we set a conservative threshold of $2 \times \sigma$ above the mean for the average of 3×3 pixels around the particle location. Here, σ denotes the background noise. We added this information in former line 242, now 236.

Reviewer #3:

6. There is a relevant work on C-iSCAT by using spinning disk confocal scanning unit which shall be added in the reference and discussed [<https://doi.org/10.1364/OE.471935>].

Our reply:

We also took note of this paper, which appeared only a few weeks ago and cites our work, which we posted on Research Square archive (Ref. 59) last May. We are happy to cite this paper now under Ref (60).

Reviewer #3:

7. The statement in Line 97 that transmission microscopy is less sensitive and has a lower axial resolution is biased. As noted by the authors (stated in the response to my comment #3), the sensitivity of interference detection is set by the number of detected photons. Small nanoscopic cell structures scatter equally in the forward and backward directions, so the detection sensitivity should be the same in transmission and in reflection.

Our reply:

Before we address the scientific part of the reviewer's comment, we point out that his/her citation of our statement on line 97 is inaccurate. We have never stated that transmission is "less sensitive". We pointed out that the contrast of a given nano-object is lower due to a larger background.

We agree that Rayleigh scattering of a nanoparticle is isotropic with respect to forward and backscattering. However, a cell content spans multiple orders of magnitude in terms of size distribution of its constituents. Mie theory shows that objects that are comparable to or larger than the wavelength radiate more in the forward direction. Consequently, imaging a cell in transmission would result in a superposition of a strong radiation from larger objects with a

weak signal due to the Rayleigh scattered light. Considering that any physical detector has a finite dynamic range, the information encoded in the intensity of the Rayleigh scattered light can be easily masked by the higher intensity signal of Mie scattering. For measurements in the reflection arrangement, on the other hand, Rayleigh back-scattered light constitutes a larger fraction of the detected light. Please also see our reply to comments below.

Reviewer #3:

The fact that most transmission microscopes reach a lower sensitivity is because they are operated at a lower incident intensity (otherwise the illumination saturates the detector). The incident power of transmission microscopy can be increased by attenuating the transmitted beam in the detection path (as being done in several interference microscope modalities, including phase contrast, DIC microscopy and others). As a result, transmission interference microscopy can also be very sensitive (in fact, it has been demonstrated that iSCAT is equally sensitive in transmission and in reflection [<https://doi.org/10.1039/C8NR06789A>]).

Our reply:

The contrast in transmission measurements is lower since the baseline intensity is higher. This means that detecting a certain feature in transmission requires a camera/detector with a much larger dynamic range (well depth and bit depth). Indeed, it is possible to spatially filter various portions of the transmitted light to manipulate the contrast, albeit at the cost of complicating the instrumentation.

Reviewer #3:

There is also no evidence that transmission microscopy has a lower axial resolution. I suggest the authors to either remove this statement or explain it explicitly in order to avoid misunderstanding.

Our reply:

The reviewer is right in that there is no fundamental limit to the axial resolution in transmission iSCAT. In other words, the gradient of the signal caused by the Gouy phase provides a means for localizing a nanoparticle in the axial direction. However, this gradient is much weaker (extends over about one wavelength) than what one obtains in reflection mode, where the contrast undergoes a full reversal within $\frac{1}{4}$ of the wavelength. Therefore, there is no doubt that the axial sensitivity is higher in reflection mode. Nevertheless, we removed this statement since it is not central to our work.

Reviewer #3:

8. The statement in Line 100 is misleading: 'when imaging smaller sub-cellular structures, it is

advantageous to carry out iSCAT measurements in a back-scattering geometry to reduce the background.' It implies that the main difficulty of visualizing nanoscopic cell structures is the background created by the large cell structures. I believe that the issue of scattering background in cell imaging is generally more complicated. For example, as evident in the data of this manuscript, the backscattering geometry picks up strong interface reflections that preclude the detection of nanostructures nearby (this is why it is difficult to observe nanostructures above or below the highly reflective nuclear membranes in C-iSCAT). The trouble of reflective background can be effectively resolved by detection in transmission. In sum, the pros and cons of forward versus backward detections are more complicated than what being stated, and they should not be oversimplified.

Our reply:

We agree with the referee that a robust comparison of the pros and cons of forward and backward detection requires more elaborate discussions. Since this is not central to our work, we have followed the reviewer's advice and have removed the original passage around Line 95-100. Instead, we make some general statements about forward and backward scattering in the beginning of the Results Section without making absolute statements about the comparison between reflection and transmission microscopy modes.

REVIEWERS' COMMENTS

Reviewer #3 (Remarks to the Author):

The authors have adequately addressed my comments and revised the manuscript accordingly.

In the authors' response to my first comment, the authors did not disagree that there are previous works studying the optical response of biological samples by considering scattering through complex PSF deconvolution. Indeed, from a methodology perspective, whether scattering is considered in an imaging method should be determined by the ways of modeling in image data processing, not by the size of the object being detected. Thus, I still consider the current manuscript underappreciates the previous efforts in modeling scattering in the phase microscopy.

Having said that, I recognize and appreciate the significance of this study in label-free nanoscopic cell imaging. I am convinced that it will make a broad impact to the community and am happy to recommend its publication in Nature Communications.

RESPONSE TO REVIEWERS' COMMENTS

Reviewer #3 (Remarks to the Author):

The authors have adequately addressed my comments and revised the manuscript accordingly.

We thank the reviewer for his/her comments and appreciation of the implemented modifications.

In the authors' response to my first comment, the authors did not disagree that there are previous works studying the optical response of biological samples by considering scattering through complex PSF deconvolution. Indeed, from a methodology perspective, whether scattering is considered in an imaging method should be determined by the ways of modeling in image data processing, not by the size of the object being detected. Thus, I still consider the current manuscript underappreciates the previous efforts in modeling scattering in the phase microscopy.

We thank the referee for revisiting this point. We have verified that we cite all relevant papers (including the one suggested in the referee's original comment) and that we do not make any misleading statement. We deem further technical elaborations about previous modelling efforts inappropriate since they would divert from the main focus of the manuscript, especially since we do not claim any novelty regarding modelling.

Having said that, I recognize and appreciate the significance of this study in label-free nanoscopic cell imaging. I am convinced that it will make a broad impact to the community and am happy to recommend its publication in Nature Communications.

We express our gratitude to the reviewer for all his/her comments, which have helped us improve our manuscript.